# Constraints and Consequences of Online Teaching

**Ciprian Obrad** 

Department of Sociology, West University of Timișoara, 300223 Timisoara, Romania; ciprian.obrad@e-uvt.ro;
Tel.: +40-723-525-454

**Abstract:** In Romania, like in many other European countries, schools and universities were closed and classes were transferred entirely online at the beginning of March 2020, due to the new coronavirus SARS-CoV-2 (COVID-19) outbreak, declared as a pandemic by the World Health Organization (WHO). In the context of drastic changes and uncertainty, teachers across Romania had to face the challenges of transferring lectures online. The small window of time left to adapt to new technologies, along with other stressors, called into question their work engagement and performance, as key factors for the sustainability of the educational system. An approach based on the dimensions of induced stress, resilience behavior, professional support, and work engagement was implemented to highlight the impact of socio-professional changes during the COVID-19 on the activity of 400 teachers from Romania. The designed construct was validated and subsequently a model was proposed, by using factor analysis and Structural Equation Modeling (SEM). The article brings updated information on the complex relationship between stress and resilience in the case of employees from the area of education. Among other results, the present research highlights the structure of work engagement and the mediating role of professional support in the relationship between stressors and resilience mechanisms.

**Keywords:** teaching; sustainable education; educational challenges; stress; work engagement; resilience; factorial analysis; SEM; covid-19

## 1. Introduction

The main purpose of the present study is to find out to what extent the Romanian teachers are able to meet the new requirements of their professional activity and if the current social state might influence their work implication. The social context we are referring to is determined by the social changes that led a series of changes and limitations, both at the psycho-social level (e.g., threats to health and safety, social distancing and isolation, etc.), and at the professional level (e.g., shifting the traditional way of face to face teaching to online education).

This research focuses mainly on the following: the changes in the teachers' working conditions, the use of new online teaching tools, the way in which the professional community of educators in Romania manages and adapts in order to attain the didactic objectives. The research project started from the assumption that the stressors of Romanian teachers are mainly related to major changes in the form of the professional act (shifting from mainly off-line education to exclusively online education as well as challenges of new educational instruments. Such factors are added to the already existing constraints of the education system. I investigated the way in which contextual factors determine the well-being of school and university employees and the way these factors are managed, in connection with work engagement. The sustainability of an education system is also achieved through the quality of the educational act, with teachers as main characters.

The research focused on identifying the main educational constraints as well as the contextual stressors, to identify the level at which the resilience mechanisms were triggered, to determine the factors that mediate resilience and work engagement. Work engagement and the current state of the

teachers in the gear of relations with other research dimensions were considered necessary conditions for ensuring the quality of the educational process in relation to the Sustainable Development Goal (SDG4) of the United Nations, created to guarantee embracive and unbiased quality education and to encourage durable learning opportunities for everyone. I also asked for the opinion of teachers from Romania, on actions that need to be taken in order to ensure the educational process sustainability, in the COVID-19 pandemic context.

### 1.1. Literature Review

To build the appropriate measurement scales for the research intentions stated above, the existing literature and the similar research in the field were studied. Based on these, I defined several dimensions and indicators to propose a conceptual model.

#### 1.1.1. Constrains and Stress for Teachers (CONSS)

According to the World Health Organization (WHO), occupational stress is defined as "response people may have when presented with work demands and pressures that are not matched to their knowledge and abilities and which challenge their ability to cope" [1].

Nowadays, in the current scientific community, there is an almost unanimous consent regarding the difficulty to narrow down stressors, due to aggregate environmental influences and different ways in which each individual manages and responds to stress.

As to occupational stress, the literature has highlighted that employees are more likely to feel some factors as being stressful in specific contexts, i.e., when there are too high expectations or requirements function to their ability to complete work in good conditions, [2] when changes occur and they are not properly managed by management [3], when there is no optimal ratio between professional demands and the time to recover and rest [4,5]. Likewise, the constraints and stress involved as the result of the professional activity can also be enhanced or mitigated by a number of factors, such as relationships with colleagues and management in terms of support given, the level of collaboration between employees, the degree of employees' work engagement and the way in which they define themselves in relation to their professional goals, as a projection of self-esteem, personal self-defense mechanisms derived from particularities of their own personality.

Regarding the stress associated with the teaching profession, there are authors who believe that "many school teachers suffer from stress and burnout" [6]. More than this, some research reports point out the fact that employees in education score higher average values of stress than employees in other sectors of activity [7]. Research on education, at various levels, has shown that the sources of stress for teachers with a direct effect on the degree of work engagement are the following: lack of their students' motivations [8], differences between constrains and workload [9], lack of support from colleagues and management [10], poor working conditions [11], and extra-organizational stressors [12].

Kyriacou [13] makes an inventory of the way in which these factors act as predictors of stress for those working in education. When the stress is not managed properly, it can lead to phenomena of chronic fatigue, defined as burnout, through its three dimensions: emotional exhaustion, depersonalization, and reduced sense of personal accomplishment [14]. Recently, a cross cultural empirical study has shown that occupational stress has a significant reverse correlation with positive mental health [15]. However, there are also studies that indicate that in specific contexts and on short term activities, some level of stress can have a boosting effect on teaching activities and it can also increase job satisfaction [16].

As concerns the measurement of stress, there are several known instruments and scales adapted in various research contexts. The most used ones are: Maslach Burnout Inventory [17], Maslach Burnout Inventory-Educators Survey (MBI-ES) version [18], Copenhagen Burnout Inventory [19] or The Perceived Stress Questionnaire from Levenstein et al. [20].

In order to measure the action of stressors in the teaching activity, in the current context, which also contain the component of constraints and stressors associated with online education

(CONSS), our own scale has been built. However, the validation process imposed the limit of this dimension to four indicators.

### 1.1.2. Negative Affects related to Stress (NEGAFF) and Positive Affects (POSAFF)

There have been a lot of attempts to show how individuals assess changes that occur in their immediate environments and the significance assigned to them. If changes or events occur and are perceived as uncontrollable, they generate stress and negative emotions [21]. Some researchers have agreed that there is a close link between stressors and negative affects, which individuals develop as a result of these, i.e., discomfort, stress, anxiety, depression [22–24]. Hu and Gruber [25] defined negative affects as being a universal outcome of unpleasant experiences and subjective stress. Negative emotions are classified as high-arousal affects, such as tension, frustration, and low-arousal affects, such as sadness and boredom. [26]

According to the existing literature, affects or positive emotions, such as joy, excitement, enthusiasm, are classified into two distinct categories: state positive affects (short-term) and trait positive affects (long-term), depending on their degree of stability, and their impact on human behavior [27]. Some researchers [28,29] show that negative emotions can also play a constructive role in learning and solving tasks. Similarly, more recent approaches [30,31] have shown that stress can be approached differently, not only through the negative emotions it causes. Positive emotions play a significant role in activating resilience mechanisms. Emotions have three dimensions: valence (positive and negative), objectifying (results), and degree of activation (they can be activators or inhibitors). Hence, there are emotions such as anger, frustration, anxiety, which are negatively activated, and there are negative emotions, such as sadness and hopelessness, which deactivate [32].

Regarding the measurement of negative and positive affects, the literature shows that there have been several scales over time: Positive and Negative Affect Schedule (PANAS) [33], the negative and positive affect scale (NAPAS) [34], or the State-Trait Emotion Measure (STEM)—scale constructed to asses emotions at the workplace [35]. To measure negative affects, I used a measurement scale with six items, out of which three items were validated.

### 1.1.3. Teachers Resilience and Coping (RESIL)

Resilience is mainly defined as a positive response to conditions of adversity [36]. The factors that determine an individual's resilience defined as protective factors of stress, can come from within (specific skills such as problem solving, a high degree of autonomy and organization through goals and future plans, some personality traits, a definite belief system such as internal locus, etc.), or be external (groups, community, social context) [37].

In the organizational and professional environment, individual resilience was analyzed in relation to work performance. It was considered a reducer of counterproductive behaviors through stress management at work (Avery et al., 2011) [38]. Resilience in the professional environment was linked to organizational commitment and work engagement behaviors. There are also authors who consider that resilience is a value that can be part of an organization's; in this case, it acts as a collective protective barrier and has the effect of an organizational environment with low stress [39,40]. There are some authors who point out that some individuals have an internal predisposition to be more or less resilient (resilience as a trait), while others claim that resilience can be learned and practiced, and use the term emotional fitness (resilience as a developing skill). Resilience, either internal/external or predisposition/behavior, becomes an individual or organizational resource that acts as a buffer against stressors [41–43]. Resilience is not is not confined to defense mechanisms triggered by adversity. Recent research has shown that there are positive conditions that can shape resilience and which have the effect of increasing the resilience of employees in organizations. Organizations support the capacity of their members to adapt to challenging circumstances through organizational culture, through stress management, through professional support, etc. Borman and Overman [44] show that resilient students are most likely to come from schools with a positive culture of teacher-student relationship

and who have grown up in supportive families. Other recent research has analyzed the relationship between stressors occurring in teachers' activity and coping strategies based on problem solving or those focused on emotions [45–48].

To measure the coping level and to identify the various strategies individuals use to respond to stressors several scales have been designed. They have different sizes, such as: scales with 52 items—the COPE Inventory in 1989 [49], or the shorter version, entitled the Brief COPE Inventory, with 28 items [50], up to even more concise versions: Coping Self Efficacy Scale (CSES) 2006, with 26 items [51], or the Brief Resilient Coping Scale (BRCS) [52].

In this article, I measured the coping strategies and the resilience of teachers using a scale, which was initially built with seven items, six of them were validated through factorial analysis.

### 1.1.4. Organizational and Professional Support (SUPP)

Next, we are going to study the role of support as a major resource of an organization. Bakker and Hakanen [53], showed that some job resources such as support from colleagues, support from the manager, and permanent information for an innovative adaptation are positively associated with an engagement in professional activity. The support that employees receive can take many forms, starting from sharing the best ways of problem solving, giving examples of good practice, all based on reciprocity, help and understanding from both the manager and work colleagues. In 2004, Schaufeli [54] referred to the two existing dimensions in each professional activity: resources and demands. These variables are closely related and can generate positive or negative results function to the dominance of one component over another and function to the specifics of the professional activity that is carried out. Thus, the result can be burnout or work engagement. Griffith et al. [55] showed that the lack of support from others (colleagues, managers) in the work of teachers becomes a stressor, while a supportive and collegial environment at work creates important premises for a serious construction of resilience and coping mechanisms. Scherer suggested that in order to adapt either to the workplace or to new activities or tools, teachers need guidance from teachers with more work experience, who are willing to collaborate [56]. Levine recognized the value of collaborative work in producing optimal results in a professional activity [57]. Other research revealed that environments that are not supportive can affect teacher job satisfaction and become stressors [58].

The measurement of the professional support that the teachers received in the educational activities was made through a scale with three items: getting help and understanding from school management, getting help from local non-governmental organizations (NGOs) and professional associations, and getting help from county and national institutions.

### 1.1.5. Work Engagement (WENG)

Work engagement is a concept that has often been analyzed in the literature as having an important role in maintaining employee health and organizational performance. On the long run, as far as teachers are concerned, work engagement becomes a prerequisite for the long-term sustainability of any education system.

There have been many ways to measure work engagement. On one hand, there are some authors sharing the idea that engagement of employees is defined in relation to burnout, the first concept being the opposite of the second one and that both can be measured with the same scale. On the other hand, there are the authors arguing that work engagement is a construct that involve a distinct measurement with a separate scale [59]. Some of the most frequently used tools for measuring work engagement, and derived from the two perspectives above, are the Maslach Burnout Inventory (MBI, MBI-General Survey (MBI-GS), which supports the measurement of work engagement using inverse scores on the scale, and the Utrecht Work Engagement Scale (UWES) [60])—which is a scale specifically dedicated to work engagement with several versions. Depending on the number of indicators included, UWES measures work engagement by three dimensions: vigor, dedication, and absorption. Vigor defines on the one hand the sustained desire to overcome obstacles that may occur at work place or in the

exerted profession and on the other hand the determination to be resilient when professional or work-related constraints and difficulties occur [61]. Dedication is the component that is activated when someone perceives his/her work as having a high level of importance for him/her and for the others. Absorption refers to the difficulty of the individual to detach himself from his own work [60]. There are also researchers who use Gallup Q12, a tool called "satisfaction-engagement" [62] that measures the engagement defined by both dimensions. Many studies have shown that work engagement occurs when the individual is fully cognitively, emotionally, and physically involved in the profession and work they do [63]. Work engagement was explained and related to other concepts, such as: emotions and affects [64], personality dimensions [65], role performance [66], job resources [67], organizational commitment [68], and job satisfaction [69].

For the purpose of this article, a scale with 10 items was created to measure the work engagement of teachers in the context of online teaching activities (scale of online work engagement in education (OWEE)). Moreover, 6 out of 10 indicators were kept after running exploratory factor analysis.

### 1.1.6. Analyzed Indicators and Implications for the Sustainability of Education

In recent approaches there is a conceptual distinction between education for sustainable development (ESD), sustainable education (SE), and other related concepts [70]. Irrespective of the differences between these concepts, the formation of a sustainable world begins with the educational process and education is a tool for a better future [71]. A lot of research has highlighted that the path to the sustainability of societies is influenced by the quality of the education systems and by the way in which they will adapt to values related to sustainable development [72]. The future of society and of the environment depends on what education systems will look like in the future [73]. Teachers have an important role in education towards sustainable development [74] and their role is crucial in ensuring the degree of professional expertise in a society in which they have the quality of role models [75,76]. The way in which teachers are engaged in teaching and the skills they acquire become pillars for the sustainability of education.

The literature has shown that the well-being of teachers, the level of stress and the length of the period during which stressors act, the conditions that ensure the work engagement (professional and organizational support), influence the quality of the educational process, which is one of the premises for ensuring sustainability in education [77–79].

### *1.2. Conceptual Model and Hypothesis*

Based on the findings resulting from the literature review, I proposed the initial conceptual model shown in Figure 1.

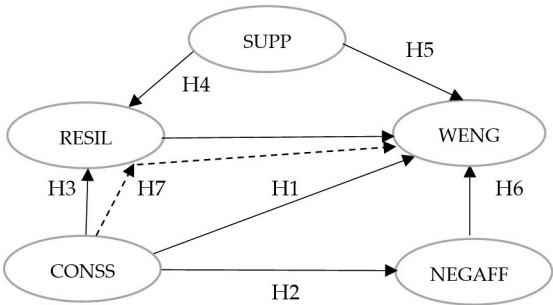

**Figure 1.** Proposed Conceptual Model.

The model postulates several assumptions starting from previous research. On one hand, the changes that took place in the educational process as a result of the COVID-19 pandemic (e.g., adaptation to new technologies, various additional professional requirements) have led to

negative emotions (e.g., loss of hope, fears). Job constrains, stress, and negative affects have a negative impact on teachers' work engagement.

On the other hand, resilience mechanisms to stressors have been triggered. Such mechanisms were enhanced by the professional support that educators received during school closure. The coping behaviors and supporting work context are positively associated with job engagement.

Following related hypotheses provided the basis of the conceptual model:

**Hypothesis 1 (H1).** *Stress (CONSS) has a negative effect on job engagement (WENG).*

**Hypothesis 2 (H2).** *There is a positive association between stress (CONSS) and negative affects (NEGAFF).*

**Hypothesis 3 (H3).** *Stress (CONSS) is negatively associated with resilience (RESIL).*

**Hypothesis 4 (H4).** *Support (SUPP) is positive associated with resilience (RESIL).*

**Hypothesis 5 (H5).** *There is a positively relationship between job engagement (WENG) and support (SUPP).*

**Hypothesis 6 (H6).** *There is a negative relationship between negative affects (NEGAFF) and work engagement (WENG).*

**Hypothesis 7 (H7).** *Resilience (RESIL) moderates the relationship between stress (CONSS) and work engagement (WENG).*

## 2. Materials and Methods

### 2.1. Sample and Data Collection

The research was conducted through online data gathering. The proposed questionnaire was developed and delivered by the Romanian platform isondaje.ro. All data were collected between 24 May and 3 June, 2020. A total of 400 out of 428 filled questionnaires were validated, after checking the compliance with the structure of the total population of Romanian teachers (distributed on 4 levels of education: primary, gymnasium, high school, and university). The existing statistical data (National Institute of Statistics Romania (INSSE-RO) *INSSE, RO*) [80] was used as a reference basis for the calibration of the sample. The 400 respondents came from 35 of the 41 counties of Romania (Table 1).

**Table 1.** Research Data Sheet

| | |
|---|---|
| **Study Universe** | 234.647 Romanian Teachers |
| **Geographical Scope** | Romania, 41 counties |
| **Data Collection Method** | Structured questionnaire distributed to professors via internet link/email |
| **Sample Unit** | Teachers in all forms of education (primary school, gymnasium, high school, university |
| **Sample** | 400 teachers |
| **Margin of Error (Confidence Interval)** | ±4.9% |
| **Confidence Level** | 95%; $z = 1.96$; $p = 0.5$ |

Source: Author.

### 2.2. Instruments and Measures

To build the measurement scales appropriate to the research intentions, the existing literature and the similar research in the field were studied. Based on these, I defined several dimensions and

indicators to propose a conceptual model. A questionnaire was designed as a research tool and it was disseminated to teachers in Romania from almost all educational cycles (primary, secondary, high school and university). The questionnaire included several measurement scales that were built and validated for the Romanian context, on a sample of 400 teachers from Romania.

An inventory was proposed consisting of measuring scales of 48 indicators grouped conceptually in 7 dimensions (Appendix A), as follows:

1. the dimension of changes at professional level—PERSCHANGE (6 indicators, referring to the impact of changes in the current socio-professional context: familiarization with the tools and platforms used in online teaching activities, adaptation of teaching content for this type of education, etc.);
2. the dimension of the constraining conditions and stress generators—CONSTRESS (10 indicators, which referred to the stress determined by the use of new technologies, rapid professional change, demotivation, exhaustion, frustration, etc.);
3. the dimension of negative affects and burnout symptoms—NEGAFF (6 indicators measuring the presence of some emotional states as: loneliness, anxiety, hopelessness and loss of meaning in life, etc.);
4. the dimension of positive affects—POSAFF (4 indicators including feelings such as love, joy, hope, etc.);
5. the dimension of resilience and coping behaviors—RESIL (it initially contained 7 indicators, referring to spending time for pleasant habits and hobbies, outdoor walks, sports, future plans, etc.);
6. the dimension of work engagement—WORKENG (10 indicators, including enthusiasm, creativity, help for pupils/students, professional fulfillment, etc.);
7. the dimension of the socio-professional help received by teachers—SUPPORT (5 indicators referring to family, friends, school or university management, NGOs and institutions with educational support).

All provided items were measured on a Likert type scale, with 5 points of intensity. The scales used for the 7 dimensions were:

(a) 1. Never, 2. Rarely, 3. Occasionally, 4. Often, 5. Nearly always;
(b) 1. To a very small extent, 2. To a small extent, 3. To a moderate extent, 4. To a large extent, 5. To a very large extent.

Additionally, two open-ended questions were provided to see the motivation behind these answers, reactions to professional problems during the pandemic period and teachers' insights on the sustainability of Romanian education.

### 2.3. Multivariate Analysis

The analyses were performed with the statistical software IBM SPSS 25 and AMOS 23.

In the first stage of the empirical approach, the exploratory factor analysis (EFA) with IBM SPSS 25 was employed, in order to examine the structure and interrelationships of the variables. For each scale of the questionnaire, I performed the principal axis factoring with orthogonal rotation of factors (varimax) in order to reduce the observed variables to a minimum number of dimensions (or components) describing the maximum proportion of variance for each variable.

Secondly, the validation of the model resulting from the exploratory factor analysis and hypothesis testing was done with IBM AMOS 23.

## 3. Results

### 3.1. Exploratory Factor Analysis (EFA)

Firstly, data were checked for suitability to factorial analysis. The Kaiser–Meyer–Olkin Measure of Sampling Adequacy has a value of 0.849. According to Kaiser and Rice [81] a value of this measure over 0.7 is acceptable and above 0.8 is meritorious. (Table 2)

**Table 2.** Kaiser-Meyer-Olkin (KMO) and Bartlett's Test.

| Kaiser–Meyer–Olkin Measure of Sampling Adequacy | | 0.867 |
|---|---|---|
| Bartlett's Test of Sphericity | Approx. Chi-Square | 8796.611 |
| | Degrees of freedom (df) | 1128 |
| | Significance probability (Sig.) | 0.000 |

Together with this, the Bartlett's test of sphericity was performed (Table 1). Sig = 0.000 was obtained, which means significance (<0.05), meaning that the correlation matrix of coefficients is not an identity matrix. In conclusion, starting from the results of these tests, I can say that the data are sufficient and suitable for factor analysis.

The factor loadings show the extent in which every measured variable is reflected in latent variables [82]. The factor loadings (FLs) have been interpreted in the sense of what Hair et al. [83] have shown. According to these authors, the present model has a significant and well-defined structure.

Regarding the communalities, as measures for reliability who indicates the proportion of each variable's variance that can be explained by the retained factors, some authors recommend to keep items with communalities over 0.2 [84]. In the actual case, only variables with higher values than 0.33 as minimum cut off point were kipped.

The visual tool of Cattell—(scree plots) were used in order to anticipate the number of extracted factors (Figure 2) [85].

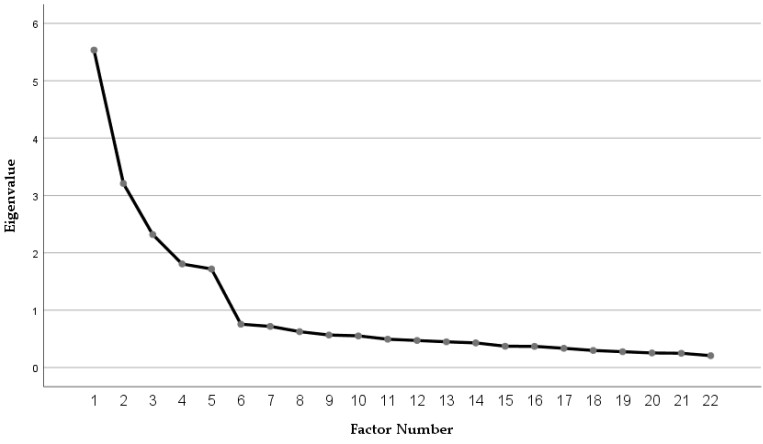

**Figure 2.** The scree plot.

Field [86] suggests that the cumulative minimum of variance explained by factors should be around 55–65%. In the present study, the 5 factors cumulative variance explained is 66.29% (Table 3)

**Table 3.** Total variance explained by 5 factors.

| Factor | Total | % of Variance | Cumulative |
|--------|-------|---------------|------------|
| 1 | 5.534 | 25.155 | 25.155 |
| 2 | 3.209 | 14.586 | 39.741 |
| 3 | 2.318 | 10.536 | 50.277 |
| 4 | 1.805 | 8.206 | 58.483 |
| 5 | 1.718 | 7.808 | 66.291 |

Another rule of decision was to follow the Kaiser rule of eigenvalues greater than one [87] and not to keep a factor with fewer than three items, because there are opinions that otherwise the factor would be too weak and unstable [88]. I eliminated 11 items out of 48, because the values of the communities were below the threshold of 0.33; other 15 items were eliminated successively because they caused cross-loadings for several factors. Thus, I achieved a reduction, and the 22 items left were extracted into 5 meaningful factors (Table 4).

**Table 4.** Factor loadings for selected indicators in explanatory factor analysis (*N* = 400).

| Indicators | Factor 1 Work Engagement | Factor 2 Resilience | Factor 3 Job Constrains and Stress | Factor 4 Negative Effects | Factor 5 Professional Support |
|------------|-------------------------|---------------------|-----------------------------------|--------------------------|-------------------------------|
| workeng 3 | 0.821 | −0.048 | −0.162 | −0.116 | 0.013 |
| workeng 4 | 0.780 | 0.036 | −0.160 | −0.100 | 0.006 |
| workeng 2 | 0.730 | 0.044 | −0.190 | −0.112 | 0.082 |
| workeng 9 | 0.710 | 0.097 | −0.037 | −0.004 | 0.040 |
| workeng 8 | 0.679 | 0.148 | −0.122 | −0.006 | 0.132 |
| workeng 5 | 0.621 | 0.096 | −0.038 | −0.015 | 0.102 |
| resil 3 | 0.075 | 0.797 | −0.038 | 0.002 | −0.006 |
| resil 7 | 0.076 | 0.778 | −0.020 | −0.036 | 0.136 |
| resil 6 | 0.040 | 0.775 | −0.112 | −0.065 | 0.197 |
| posaff 2 | 0.110 | 0.749 | −0.088 | −0.105 | 0.088 |
| resil 1 | 0.077 | 0.649 | −0.053 | 0.031 | −0.014 |
| resil 4 | 0.006 | 0.552 | −0.107 | −0.025 | 0.112 |
| constress2 | −0.230 | −0.010 | 0.815 | 0.172 | −0.074 |
| constress1 | −0.089 | −0.047 | 0.732 | 0.117 | 0.062 |
| perschange2 | −0.182 | −0.108 | 0.597 | 0.015 | −0.075 |
| constress4 | −0.104 | −0.285 | 0.586 | 0.105 | −0.184 |
| negaff5 | −0.086 | −0.078 | 0.185 | 0.853 | −0.100 |
| negaff6 | −0.077 | −0.046 | 0.062 | 0.825 | −0.019 |
| negaff4 | −0.086 | −0.022 | 0.105 | 0.755 | −0.038 |
| support4 | 0.134 | 0.074 | −0.020 | −0.111 | 0.724 |
| support5 | 0.121 | 0.157 | −0.038 | −0.001 | 0.708 |
| support3 | 0.023 | 0.119 | −0.109 | −0.026 | 0.664 |
| Variance explained | 25.15% | 14.58% | 10.53% | 8.20% | 7.80% |
| Eigen Value | 5.53 | 3.20 | 2.31 | 1.80 | 1.71 |
| Cronbach's alpha | 0.81 | 0.86 | 0.80 | 0.86 | 0.76 |

Note: workeng (work engagement), resil (resilience), posaff (positive affects), constress (job constrains and stress), negaff (negative affects), support (support).

The items kept according to all decisions presented above and that arise from the analysis needs are presented in Table 5. The dimensions containing these indicators were included in the SEM (structural equation modeling) and also in the complementary analyzes performed in the following sections of this article.

**Table 5.** Selected items of model and their measuring system.

| Factors/Items | Measuring Scale |
|---|---|
| Work Engagement/6 items<br>feeling enthusiastic about work<br>being creative in teaching online<br>being fulfilled with job<br>feeling that I helped students to learn<br>being energetic in professional tasks<br>feeling engaged in the educational process | (1—Never, 5—Nearly always)<br>Rating by answer from 1 to 5<br>for each item ascending |
| Resilience and coping behaviors/6 items<br>focusing on the good interaction with family<br>making personal future plans<br>enjoying sleep and rest<br>enjoying time for personal activities<br>involving in old activities and hobbies<br>making outdoor walks | (1—Never, 5—Nearly always)<br>Rating by answer from 1 to 5<br>for each item ascending |
| Job constrains and stress/4 items<br>feeling that everything about work is changing too quickly<br>being tensioned about online teaching<br>feeling the stress of adapting to new technologies<br>feeling exhausted related to teaching online | (1—Never, 5—Nearly always)<br>Rating by answer from 1 to 5<br>for each item ascending |
| Negative affects/3 items<br>feeling hopeless<br>feeling depressed<br>losing the meaning of life | (1—Never, 5—Nearly always)<br>Rating by answer from 1 to 5<br>for each item ascending |
| Professional Support/3 items<br>receiving support from school management<br>receiving support from local NGOs and professional associations<br>receiving support help from county level and national institutions | (1—to a very small extent, 5—to a very large extent)<br>Rating by answer from 1 to 5<br>for each item ascending |

## 3.2. Confirmatory Factor Analysis (CFA)

The validation of the model resulting from the exploratory factor analysis was done with IBM AMOS 23 soft. I imported from SPSS the rotated factor matrix of the 5 factors, together with loading factors for each item, using the plugin from Gaskin and Lim [89]. In order to validate the model, I made tests with confirmatory factor analysis (CFA). The aim of the confirmatory factor analysis (CFA) is to help with making decisions about keeping some indicators in scales and instruments, and generally when assessing construct validity [90]. The purpose of the CFA is also to obtain an estimate for each parameter for the proposed model (e.g., load factor, variance factor and covariance, indicator error variance and covariance), that produce a predicted variance-covariance matrix, which resembles as close as possible the variance-covariance matrix sample [91]. First priority was to check the reliability and the validity of the conceptual model. The degree of consistency in measurement among different repeated trials expresses the reliability of a procedure or an instrument [92]. The value of composite reliability (CR) reflects good reliability when it is greater than 0.7 [93]. Cronbach's alpha indicates the internal reliability of a construct. According to the interpretation of results by DeVellis [94], the present model has 3 "very good factors" and 2 "respectable" ones. The extent to which the collected data are accurately measuring what was intended to measure is the core of validity [95]. A model must indicate that there is evidence for both convergent validity and discriminant validity, because none of them is sufficient alone [96]. The convergent validity is reached when an indicator strongly correlates its construct. The convergent validity resulting from factor loadings shows a good fit in the model. The statistical criterion [97] for convergent validity is that any factors explain more than 50% of construct variance and when the value of composite reliability is greater than the value of the average variance extracted (AVE > 0.5 and CR > AVE). (Table 6)

Discriminant validity refers to the extent to which a set of variables are correlated with their variable outcomes. The statistical criteria for discriminant validity are that a construct should extract more variance AVE than the maximum shared variance by it with any other construct (AVE > Maximum

Shared Variance (MSV)) and secondly, the amount of variance that a latent construct extracts in comparison with the error term should be more than the average amount of variance that is shared with any other construct in the model; (AVE > ASV).

**Table 6.** Model reliability and validity Measures.

| Factors | CR | AVE | MSV | ASV |
|---------|------|------|------|------|
| 1 | 0.878 | 0.548 | 0.174 | 0.042 |
| 2 | 0.873 | 0.537 | 0.093 | 0.035 |
| 3 | 0.807 | 0.516 | 0.174 | 0.069 |
| 4 | 0.866 | 0.684 | 0.122 | 0.047 |
| 5 | 0.761 | 0.516 | 0.093 | 0.036 |

Note: CR = Composite Reliability, AVE = Average Variance Extracted, MSV = Maximum Shared Variance, ASV = Average Shared Variance.

I have established so far that the model has 22 observed variables (endogenous) grouped in 5 latent variables (factors). The hypothesized model has a $\chi 2$ value of 446,968 with 199 degrees of freedom and probability 0.000 and it shows the fact that the minimum value of discrepancy between unrestricted matrix of covariance and restricted matrix of covariance of the model has been achieved. Furthermore, an assessment for normality was done in order to continue the proposed analysis. In the assessment of normality section from AMOS output, it can be noticed that kurtosis values are positive and negative values, leading to an average value of −0.354. There are researchers [98,99] who argued that data is considered to be normal if skewness is between −2 to +2 and kurtosis is between −7 to +7 and only values over 7 indicate a distance from normal. Curran et al. [100] suggest the same moderate normality thresholds of 2.0 and 7.0 for skewness and kurtosis respectively, when assessing multivariate normality. Moreover, in the section of observation farthest from the centroid (Mahalanobis distances), I observe that differences between values are not significant, data being not distanced, with the only exception of an outlying case. The Mahalanobis distance measures the standard deviation between the individual values and the sample means for all variables (centroid) average of all values [101]. According to the estimates, the multivariate distribution is close to a normal distribution, with no serious reasons to reject the assumption of normality. In that case, further analyzes can be done in AMOS. Before testing the hypothesized structural model, the measurement model was first constructed to estimate the latent variables.

The result of performed test indicates in Table 7 shows an already good fit to the data: $\chi 2$ tests, the goodness of fit index (GFI), the comparative fit index (CFI), the root mean square error of approximation (RMSEA), the standardized root mean squared residual (SRMR), *p* of close fit (PCLOSE), and the Tucker–Lewis index (TLI). The initial values were more than acceptable, referring to Hu and Bentler [102]. The fit of the model can be improved by drawing covariances between two pairs of errors in the same construct, as suggested by high values of modification indices (MI). After drawing covariances, the results of the performed test show an improvement to already good indices, with a decrease in value of $\chi 2$ and $\chi 2/df$ and increase in *p* of close fit value (Table 7). The measurement model is presented in Figure 3.

**Table 7.** Possible improvement of model fit indices.

| Statistics | $\chi 2$ | df | *p* | $\chi 2/df$ | GFI | CFI | RMSEA | PCLOSE | SRMR | TLI |
|-----------|------|-----|-------|--------|------|------|-------|--------|------|------|
| Recommended values | | | | <3 | >0.9 | >0.9 | <0.08 | >0.05 | <0.08 | >0.9 |
| $T_0$ | 446.96 | 199 | 0.000 | 2.24 | 0.90 | 0.94 | 0.05 | 0.08 | 0.05 | 0.93 |
| $T_1$ | 390.68 | 197 | 0.000 | 1.98 | 0.91 | 0.95 | 0.05 | 0.53 | 0.05 | 0.95 |

Note: $T_0$—at the beginning, $T_1$—after drawing covariances, $\chi 2$ (chi square), df (degrees of feeedom), *p* (probability level), GFI (goodness of fit index), CFI (comparative fit index), RMSEA (root mean square error of approximation), PCLOSE (closeness of fit), SRMR (standardized root mean square residual), TLI(Tucker-Lewis index).

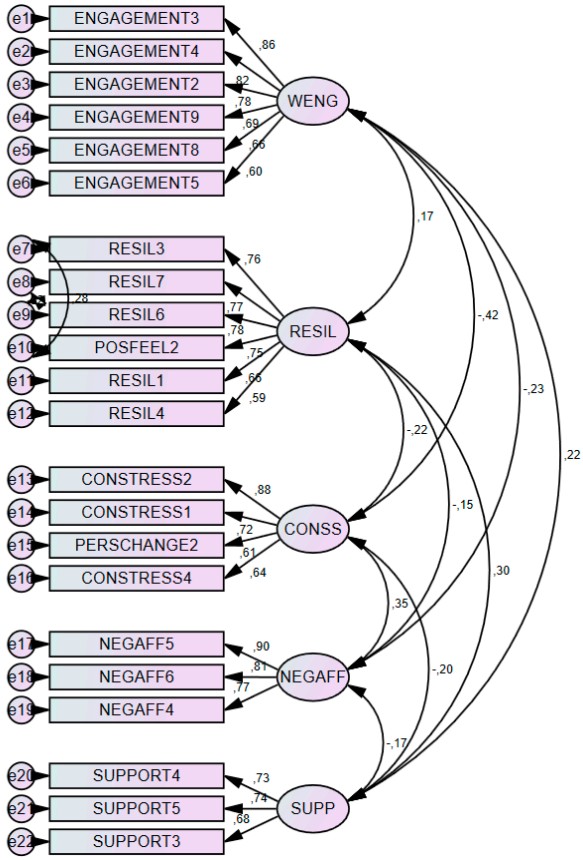

**Figure 3.** Confirmatory factor analysis (measurement model).

### 3.3. Testing Hypothesis and Structural Equation Modeling (SEM)

Furthermore, in order to examine the assumed relationships by hypothesis between job constraints, negative affects, support, resilience and work engagement, a path analysis approach with structural equation modeling (SEM) was done.

Path coefficients suggested that job constrains and stress were directly and positively associated with negative affects ($\beta = 0.36$, $p < 0.001$); job constrains and stress have a negative effect on work engagement ($\beta = -0.42$, $p < 0.001$); job constrains are negatively associated with resilience ($\beta = -0.17$, $p < 0.05$); there is a positive relationship between received support and work engagement ($\beta = 0.16$, $p < 0.05$); resilience is positively associated with support ($\beta = 0.27$, $p < 0.001$). There is no significative relationship between negative affects and work engagement ($\beta = -0.07$, $p > 0.05$) and there is no mediation of resilience between job constrains and work engagement. (Table 8)

The Bootstrap test was used to examine the significance of the mediating effects of resilience on the relationship between Job Constrains and Stress and Work Engagement. After testing direct and indirect effects, the conclusion was that no direct mediation was found. Neither the direct effect from supposed mediator to WENG is significant ($p > 0.05$), nor the indirect effect of mediation from CONSS (Job constrains and stress) to WENG (work engagement) ($p > 0.05$), so there is no real mediation. The H7 hypothesis of mediation cannot be supported.

However, form path model and bootstrap (2000 samples) test results that receive support (SUPP) partially moderate the relationship between CONSS and RESIL. After testing for all direct and indirect effect for all cases, the results show there are significant relationships ($p < 0.05$).

**Table 8.** Results of hypotheses testing for direct and mediating associations.

| Hypothesized Relationships | β | S.E. | C.R. | p | Decision |
|---|---|---|---|---|---|
| H1: Job constrains and stress → Work engagement | −0.42 | 0.05 | −7.456 | *** | Accepted |
| H2: Job constrains and stress → Negative affects | 0.36 | 0.06 | 6.295 | *** | Accepted |
| H3: Job constrains and stress → Resilience | −0.17 | 0.07 | −2.916 | < 0.05 | Accepted |
| H4: Received Support → Resilience | 0.27 | 0.07 | 4.164 | *** | Accepted |
| H5: Work engagement → Received Support | 0.16 | 0.07 | 2.378 | <0.05 | Accepted |
| H6: Negative affects → Work engagement | −0.07 | 0.04 | −1.657 | >0.05 | Rejected |
| H7: Stress → Resilience → Work engagement | CONSS to RESIL direct effect significant RESIL to WENG direct effect not significant Indirect effect from CONSS to WENG via RESIL not significant | | | | Rejected |

Note: CONSS (job constrains and stress), WENG (work engagement), RESIL (resilience), NEGAFF (negative affects), β—Standardized regression weight, S.E.—Standard error, C.R.—Critical ratio, *p*—Probability value, where *** means *p* < 0.001.

The results of testing the hypotheses are shown in Figure 4.

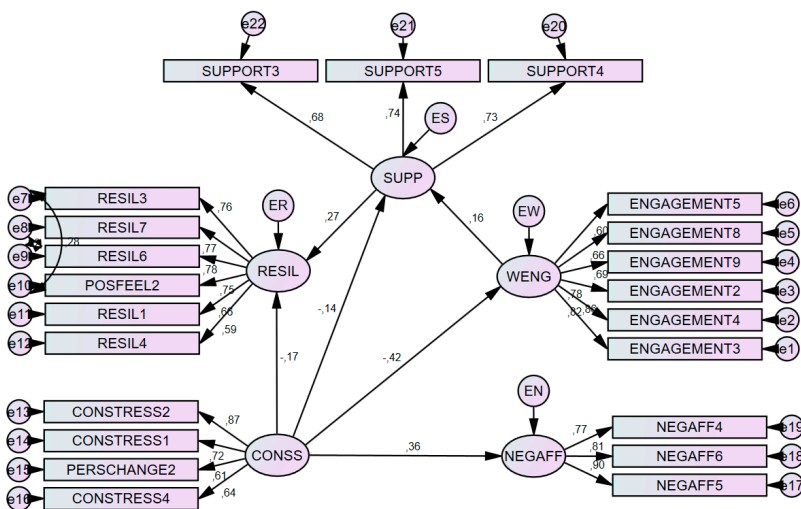

**Figure 4.** Adjusted structural equation model (SEM) with path diagrams and standardized regression weights.

### 3.4. Complementary Analysis

In order to compare the degree to which different indicators of the 5 dimensions (CONSS, RESIL, SUPP, NEGAFF, WENG) have a higher contribution to the predominance of a dimension, I have rated each answer variant as follows: 5 points for "to a very high degree", 4 points for "to a high degree", 3 points for "to an adequate degree", 2 points for "to a small degree", and 1 point for "to a very small degree". Therefore, for each indicator, regardless of assigned dimension, the maximum possible score is 2000. It results from the number of respondents (400). The maximum scale score for the answer is "to a very high degree" (5 points). Consequently, the minimum score for each indicator is 400. In order to

compare the scores obtained at the level of each indicator and dimensions, the methodological indices from Table 9 were used.

**Table 9.** Methodology score indices for each of 5 dimensions.

| Indices | Calculation Method or Formula |
|---|---|
| Maximum score of indicators ($M_{ax}SI$) = 2000 | The theoretical number if each respondent were to score 5 points for each indicator |
| Minimum score of indicators ($M_{in}SI$) = 400 | The theoretical number if each respondent were to score 1 point for each indicator |
| Maximum dimension score ($M_{ax}SD$) | MSD = $M_{ax}SI$ number of measured indicators |
| Minimum dimension score ($M_{in}SD$) | $M_{in}SD$ = $M_{in}SI$ number of measured indicators |
| Dimensional score (DS) | The empirical (cumulative) score, resulting from adding IS for each dimension |
| Indicator score (IS) | The empirical (cumulative) score, resulting from adding the answers for all respondents, for each indicator |
| Dimensional average score (DAI) | The average of indicator's score for each dimension |

Then, I divided the range between the minimum score and the maximum score of each dimension in equal distances, having 5 scale scores on a continuum from minimum to maximum. The minimum was labelled as "very low" and maximum as "very high". Thus, 4 equal intervals were obtained for each dimension, as follows: "very low to low", "low to moderate", "moderate to high", "high to very high". After that, the average for each dimension was calculated, starting from the indicators scores. Table 10 shows the values obtained and the classification for the four dimensions included in the study, as well as the indicators that had the highest/lowest scores compared to dimensional average score (DAI):

**Table 10.** Contribution of dimensions and indicators to diagnosis.

| Dimension Containing Indicators Number | Framing in Scale Intervals | DAI for Each Dimension | The Indicator Contributing the Most to DAI | The Indicator Contributing the Least to DAI |
|---|---|---|---|---|
| CONSS (4) | from moderate to high | 1340.25 | feeling the stress of adapting to new technologies | being tensioned about online teaching |
| RESIL (6) | from low to moderate | 1104.16 | enjoying time for personal activities | making outdoor walks |
| SUPP (3) | from low to moderate | 1118 | receiving support from school management | receiving support help from county level and national institutions |
| NEGAFF (3) | from very low to low | 732.66 | feeling hopeless | losing the meaning of life |
| WENG (6) | from moderate to high | 1300.5 | feeling enthusiastic about work | losing the meaning of life |

## 4. Discussion

Hypothesis 1 was confirmed and it indicates an inverse relationship between job constraints and stress (CONSS) and the work engagement of the Romanian teachers (WENG). Considering the fact that the education activity consisted only in remote teaching, the number of professional demands among those working in the educational field has increased. Among these, we can mention: the stress induced by new technologies, new tools involved in online teaching, additional time given to transposing and

adapting teaching materials for this new type of delivery, a pronounced dynamic of changes. All these changes in a context of uncertainty made teachers feel the stressors more acutely. The intensity of the stressors felt by the teachers, since the beginning of the COVID-19 pandemic, was moderate to high. All of these factors have an inversely proportional effect on the degree of work engagement. The stress generated by adapting to new technologies has the strongest demotivating effect of all the indicators.

Nevertheless, there are also indirect, contextual causes that may be associated with stressors at the professional level. For instance, in the case of teachers working from home, due to the current circumstances of the COVID-19 pandemic, they had to carry out, besides their professional activity, other responsibilities as well. Thus, they had to take over many family responsibilities, such as: quasi-full parenting or full parenting; especially when the other parent did not have a remote job.

Regarding Hypothesis no. 2, which was also confirmed, I can say that the changes that took place and the stress caused by technology led to the occurrence of negative emotions. Among them, the most common one is the loss of hope, enhanced by the idea that there are small chances that the teaching-learning activities will take place directly in front of the pupils/students. However, the intensity of the negative affects was felt at a rather low level.

Stress factors have also influenced the level of resilience behaviors which the teachers adopted in this period. The inverse proportion that exists is of low intensity; its existence being proved by the confirmation of Hypothesis 3. It is very likely that a longitudinal analysis would have brought additional clarifications about the exact moment when the coping mechanisms were triggered. The factor that best counterbalanced the effect of stressors was the larger amount of time available for teachers to enjoy personal activities, in the personal home environment together with their families.

The increase of the resilience capacity can be enhanced by supportive work contexts. This can occur when there are means to facilitate professional resources, such as the support that teachers have benefited from during this period; fact confirmed by Hypothesis no. 4 which shows that the support received during this period directly influenced the increase in resilience. The greatest contributor in this respect was the help and understanding provided by the management of the department or school/university and to a lesser extent the support that teachers received from institutions at the local or national level. The trend I noticed is the following: the greater the distance between teachers and the initiator of the support actions (local-national), the lesser the contribution of the support received to build resilient behaviors.

From a statistic point of view the confirmation of Hypothesis no. 5 leads to another finding, namely: teachers who are more engaged at work are willing to seek and employ sources of support more quickly. This proves that there is a direct relationship between professional involvement and the support received. In other words, supportive mechanisms become a professional resource that is activated when the level of work engagement increases.

One of the findings of this article was that support (SUPP) mediates the relationship between stressors and resilience mechanisms. There have been other recent studies that have highlighted the existence of the mediating role of support between stress and life satisfaction [103]. The novelty of this article is that it brings updated information about the complex relationship between stress and resilience in the case of employees in education field. Teachers who have high levels of stress may maintain low levels of support, and have low resilient behaviors. Besides, the perceived professional support has a role in increasing teachers' resilience to stressors.

In addition to the main stress factors during this time, some positive effects of the current educational context were also identified. Although schools have moved online and the physical distance between teachers and students has increased, the psychological distance has decreased, and teachers have felt that they are more connected to their pupils/students. The idea of being affected together by the same transitions and challenges acted as a social glue, in the sense that Godwin said in 2005, about the importance of the social relationships that help human individuals cope with uncertainty in their changing world [104]. Further research can be developed based on this variable in order to explain work engagement.

Beyond analyzing how the relationships between the above mentioned dimensions with their corresponding indicators are related in the research (stress, resilient behaviors, work engagement and professional support), two open-ended questions were introduced in the questionnaire applied to teachers. The purpose of the research was to reveal not only the main problems and concerns determined by the pandemic context to which teachers are trying to find solutions, but also immediate interventions that teachers consider vital to ensure the sustainability of the Romanian education system.

The study identified that one of the main problems of the moment for which teachers in Romania are trying to find solutions is keeping their health and that of their family members within the context of corona virus pandemic. This concern is caused by the risk of contamination and by uncertainty; 40%of the respondents invoked this aspect. Another concern for almost a quarter of teachers is connected to the issue of education: what will happen during the next school year, how will be the teaching activity, online teaching, assessments and the situation of their pupils and students. The third issue in the order of importance, in terms of the intensity felt, is caused by the lack of the ordinary socialization, typical before the pandemic, the recommencement of social life and social interactions (15%).

Regarding the necessary actions for a sustainable education in Romania in the immediate future, 20.7% of the teachers participating in the survey invoked the need for increased funding for the Romanian education. This means to modernize it, to grant access to a high-performance technology and to digitalization. Many of those who support this intervention believe that this would increase their motivation and work engagement. It is noteworthy that the percentage of Gross Domestic Product (GDP) assigned to education in Romania is among the lowest in the European Union (3.2% compared to a European average of 4.6%) [105]. Another priority intervention in the opinion of 17.6% of Romanian teachers, consists in equalizing opportunities in education through access to technology and internet. Remote education through online technology has deepened, as in the rest of the world, the inequalities the access to education of the children from disadvantaged families and poor areas. 40% of the Romanian children are at the limit of the risk of social exclusion, over a half of them living in rural areas, where the Internet has a penetration rate of only 47% [106]. This idea is in line with the United Nations SDG4 on ensuring inclusive and equitable quality education and promoting lifelong learning opportunities for all [107].

Another practical solution that contributes to the sustainability of education in Romania involves the imperative of improving teachers' digital skills by participating to training courses (9.2%). They are necessary to more easily manage constraints caused by the context of school closure and transition to remote education. Adaptation to new technologies is one of the indicators that generates highest level of stress, and the lack of skills in this regard reduces the engagement in the professional activity in the long run, as we noticed in the analysis presented above. Recent research has also shown that acquiring and improving digital skills is a condition of the quality of education, as indicated by United Nations sustainable development Goal 4 [108].

Teachers have also identified other solutions that can ensure the sustainability of Romanian education, including: reforming the education system through a new paradigm aimed at reorganization and modernization (10.2%), revision of school curricula, teaching objectives and taught content to provide an education anchored in practice and adapted to the current context and needs of society (8.9%). Education should be the basis for the sustainable development of a society undergoing major change. There have been other recent studies that have highlighted the need for the education system to adapt quickly to changes and to promote itself the change in accordance with the societal context [109].

Approximately 8.7% of the teachers in Romania believe that the viability of education depends on the return to face to face teaching and on the reopening of the schools.

The data analysis also showed that higher education teachers had a higher adaptability response than primary or secondary school teachers, due to the greater freedom they had in using platforms or tools for online teaching. Work engagement and professional support were more evident in this type of education. There was also a higher success rate in urban schools, regardless of the type of education (primary, secondary or tertiary). Beyond the stress of adapting to new online teaching technologies,

another problem felt acutely in terms of the educational process was related to the impossibility of optimal control in the evaluation of the pupils and students and of the feedback received from them.

The article provides a complex perspective of the relationships that take place between factors of occupational stress, professional support, resilience and work engagement of the teachers in Romania. Reducing high-stress demands that can have negative affects can increase the work engagement. By activating and increasing some professional resources, such as: supporting teachers to improve their skills in the use of new technologies involved in online education and development, the resilience mechanisms can be triggered more readily.

The results of the study can be used for various intervention mechanisms to increase the well-being of the teachers and their level of work engagement; essential conditions for the sustainability of the education system.

The purpose of the educational process in general, is to be able to take place in a context in which stress levels are not too high and in which there is an increased level of satisfaction of all the stakeholders. One of the main sustainability goals (goal number 4) of the 2030 Agenda for Sustainable Development by the United Nations focuses on the equal rights of all to receive a fair and quality education. In this regard, teachers play a crucial role, because a sustainable education involves "teachers engaged and empowered, motivated and supported within well-resourced, efficient and effectively governed systems" [110].

The sustainability and the quality of education can also be defined by the impact on the final school year results, translated by engagement, quality of life of future graduates and also by the well-being of teachers in their role as front-runner educational stakeholders. The teachers' stress affects the success of those who learn from them, just as a better engagement in work activities of people involved in the education system offers greater chances for the teaching activities to lead to optimal results.

In the current context, teachers have had to change on the go the ways in which the contents of the teaching reach their pupils and students. In the future, it is likely that there will be a need to also change the actual content, what is transmitted through the educational process, so that we can talk about the sustainability of education systems as the teachers' ability to adapt their content to the challenges and changes of the present context.

The study has a limit, given by the fact that the research is a cross-sectional one, meaning that it is difficult to establish a causal determinism between the presented variables. The study can be completed by a longitudinal analysis to capture the cause-effect relationship.

The article offers a perspective on the relationship between analyzed variables in the Romanian education for the period of few months that has passed since the COVID-19 pandemic was declared. Due to the existing uncertainty and the possibility that this context is prolonged, further research is needed to add clarifications in the long run concerning the impact on the sustainability of education.

**Funding:** This research received no external funding.

**Acknowledgments:** The author appreciates the collaboration from teachers all over Romania that responded to the survey, providing valuable information for this paper. The author would also like to thank the anonymous experts and reviewers for their valuable suggestions.

**Conflicts of Interest:** The author declares no conflict of interest.

## Appendix A

**Table A1.** Initial measuring scales and indicators.

| Dimensions/Indicators | Measuring Scale |
|---|---|
| Perschange/6 items<br>spending time for familiarization with the online tools<br>spending time for adaptation of taught content for<br>online education<br>struggling with uncertainty in professional activity<br>making effort for students to actively participate to online meetings<br>making effort to evaluate students<br>feeling compelled to compromise on education | (1—Significantly decreased, 5—Significantly increased)<br>Rating by answer from 1 to 5<br>for each item ascending |
| Job constrains and stress/10 items<br>feeling that everything about work was changing too quick<br>feeling detached from work<br>feeling frustrated with online teaching<br>being demotivated about job<br>feeling mentally tired<br>feeling exhausted in activity<br>having insomnia<br>feeling like crying<br>feeling tensioned about online teaching<br>feeling to have too much to do and too little time | (1—Never, 5—Nearly always)<br>Rating by answer from 1 to 5<br>for each item ascending |
| Negative affects/6 items<br>feeling alone<br>being stressed<br>feeling nervous<br>feeling depressed<br>feeling hopeless<br>losing the meaning of life | (1—Never, 5—Nearly always)<br>Rating by answer from 1 to 5<br>for each item ascending |
| Positive affects/4 items<br>feeling gratitude for having my family around<br>having hope that things will change for better<br>feeling the pleasure of having extra time for me<br>feeling the joy of nature and beautiful weather | (1—Never, 5—Nearly always)<br>Rating by answer from 1 to 5<br>for each item ascending |
| Resilience and coping behaviors/7 items<br>focusing on good interacting with family<br>making personal future plans<br>enjoying sleep and rest<br>enjoying time for personal activities<br>involving in old activities and hobbies<br>making outdoor walks<br>doing sport and physical activities | (1—Never, 5—Nearly always)<br>Rating by answer from 1 to 5<br>for each item ascending |
| Work Engagement/10 items<br>creating a positive atmosphere among the participants in the<br>online activities.<br>feeling enthusiastic about work<br>being creative in teaching online<br>being fulfilled with job<br>feeling that helped students to learn<br>being energetic in professional tasks<br>feeling engaged in the educational process<br>feeling connected with students<br>feeling passionate about work<br>feeling like a role model for students | (1—Never, 5—Nearly always)<br>Rating by answer from 1 to 5<br>for each item ascending |
| Professional Support/5 items<br>receiving support from my family<br>receiving support from students and their families<br>receiving support from school management<br>receiving support from local NGOs and professional associations<br>receiving support help from county level and national institutions | (1—To a very small extent, 5—To a very large extent)<br>Rating by answer from 1 to 5<br>for each item ascending |

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
