# Peer review of "Constraints and Consequences of Online Teaching"

_sustainability, doi:10.3390/su12176982_

Round 1
Reviewer 1 Report
I enjoyed reading this submission. It is topical and timely.
The presentation is thorough and acceptably clear to a reader who is not primarily a statistician.
Some proofreading is needed - mostly for spelling. The word "certain" is used with perhaps too great a frequency, where synonyms (e.g. particular) or examples (such as...) might be used instead. However, I found the paper quite readable.
I was surprised that, having mentioned two open-response questions on the instrument, there did not appear to be any analysis of the data they provided.
I was also a bit surprised at the lack of discussion of the rejected H6.
A key for f1 would be helpful (to remind the reader of the abbreviations and acronyms), and it would also be useful to label the hypotheses on the diagram (is H5 shown?!).
There is a note (line 253) that a representativeness error has been estimated - what statistic was calculated?
Line 523 - I am not sure, by the way, that I completely agree that education should take place without stressors - line 89 and ref 16 indicate that some level of stress is likely to be beneficial to performance and job satisfaction.
Thank you.
Author Response
I made changes recommended by reviewers as follows:
Comments and Suggestions from Reviewer 1 |
Detalis of the revisions |
The word "certain" is used with perhaps too great a frequency, where synonyms (e.g. particular) or examples (such as...) might be used instead. |
I have replaced the word “certain” with synonyms |
I was surprised that, having mentioned two open-response questions on the instrument, there did not appear to be any analysis of the data they provided. |
I analyzed the answers to the open-ended questions and added them to the final part of discussions |
I was also a bit surprised at the lack of discussion of the rejected H6. |
I presented the decision taken and consequently the rejection of hypothesis H6(now hypothesis 7) (lines 506-511 of article); “The Bootstrap test was used to examine the significance of the mediating effects of resilience on the relationship between Job Constrains and Stress and Work Engagement. After testing direct and indirect effects, the conclusion was that no direct mediation was found. Neither the direct effect from supposed mediator to Weng is significant (p > 0.05), nor the indirect effect of mediation from CONSS (Job constrains and stress) to WENG (work engagement) (p > 0.05), so there is no real mediation. The H6 hypothesis of mediation cannot be supported.”
|
A key for f1 would be helpful (to remind the reader of the abbreviations and acronyms), and it would also be useful to label the hypotheses on the diagram (is H5 shown?!). |
I redrawed the diagram, adding H5 and another missing hypothesis. I have also labeled the hypothesis on the diagram |
There is a note (line 253) that a representativeness error has been estimated - what statistic was calculated? |
I changed the “representativeness error” term with a more appropriate one: margin of error (confidence interval) which refers to the level of precision of the sample relative to the total population. I chose the value of z from z scores tables (z = 1.96 assuming a 95% confidence level and a 0.5 standard deviation). I had used the Cochran’s Sample Size Formula which I considered appropriate because my study population is a large population |
Line 523 - I am not sure, by the way, that I completely agree that education should take place without stressors - line 89 and ref 16 indicate that some level of stress is likely to be beneficial to performance and job satisfaction. |
I reprashed the line: “The purpose of the educational process in general, is to be able to take place in a context in which stress levels are not too high” in order to align with the idea that some level of stress can have a boosting effect on teaching activities and also can increase job satisfaction (mentioned in introductory section)
|
Thank you for your time and for your help to improve this article!
Sincerely,
Ciprian Obrad, West University of Timișoara, Department of Sociology
Reviewer 2 Report
Congratulation. Excellent work!
Please rephrase following lines from your document, as identical citing was detected for citing some of your sources:
77-78 when citing [7]
87-90 when citing [16]
346-348 when citing [80]
440-442 Therefore …for the answer
I assume it is not plagiarism, but it could count as such, so I recommend rephrasing these small parts, in order to value the excellent work of the whole paper.
Author Response
I made changes recommended by reviewers as follows:
Comments and Suggestions from Reviewer 2 |
Detalis of the revisions |
Please rephrase following lines from your document, as identical citing was detected for citing some of your sources: |
I rephrased those lines as you suggested
Lines 86-89 Other researchers state that, teachers experienced lower job satisfaction when compared to other highly stressed occupational groups replaced with Some research reports point out the fact that employees in education score higher average values of stress that employees in other sectors of activity
Lines 99-103 There are also studies that have shown that a certain level of stress can have a positive effect on the effectiveness of the teaching process and it can lead in the long run to an increased level of job satisfaction
Replaced with Other authors have shown that in specific contexts and on short term activities, some level of stress can have a boosting effect on teaching activities and also can increase job satisfaction.
Lines 427-431
The use of CFA to investigate the construct validity of hypothesis-based testing instruments adds a level of statistical precision and can assist in the development of abbreviated forms of an instrument or confirmation Replaced with The aim of confirmatory factor analysis (CFA) is to help with making decisions about keeping or not some indicators in scales and instruments and generally when assessing construct validity
I rephrased the paragraph from lines 526-531 Therefore, for each indicator, whatever the dimension to which it is assigned may be, the maximum possible score is 2000. It results from the number of respondents (400). The maximum scale score for the answer is “to a very high degree” (5 points).
|
Thank you for your time and for your help to improve this article!
Sincerely,
Ciprian Obrad, West University of Timișoara, Department of Sociology

Reviewer 3 Report
The abstract is too narrative and should focus on objectives and results from the research itself. In addition, the research cannot be an approach. May be, authors want to stress that “An approach based on the dimensions of induced stress, resilience behavior, professional support and work engagement was implemented to highlight the impact of socio-professional changes during the Covid19 on the activity of teachers”.
The introduction is confusing. There is no reference to Sustainability or to Educational sustainability development. I addition, the link between the internal conditions for teaching and the dimensions of analysis are not fully established. Lines 49 to 52 should be moved to the methodology section.
Introduction is poor in putting on context the dimensions of the study in the Higher Educations Institutions and specially, in teaching.
The conceptual model of Figure 1 should largely be described, and linked to hypotheses.
I am missing the indicators for each dimension in page 7.
For numbers, use appropriate SI units, with points for decimals.
Rephrase ‘meaning that my correlation matrix of coefficients is not an identity matrix.’. The correlation matrix is not yours. Correct also ‘I can say that my data are sufficient and suitable for factor analysis.’ You cannot say: my data. Whenever possible, do not use the possessive ‘my’: my model, my data, etc…
In the whole manuscript I am missing references on sustainability. The author should make an effort to establish a relationship between the final 22 indicators and sustainable development in primary, secondary and tertiary education. The relationship between Goal 4 and the present study is not established in the model.
In the discussion, remove: As I pointed out in the theoretical part of this article.
The discussion is poor and should be fully referenced with references from the educational world.
This study should include a final paragraph on limitations.
Author Response
I made changes recommended by reviewers as follows:
Comments and Suggestions from Reviewer 3 |
Detalis of the revisions |
The abstract is too narrative and should focus on objectives and results from the research itself. In addition, the research cannot be an approach. May be, authors want to stress that “An approach based on the dimensions of induced stress, resilience behavior, professional support and work engagement was implemented to highlight the impact of socio-professional changes during the Covid19 on the activity of teachers”. |
I have revised the abstract and change it as you suggested |
The introduction is confusing. There is no reference to Sustainability or to Educational sustainability development. I addition, the link between the internal conditions for teaching and the dimensions of analysis are not fully established. Lines 49 to 52 should be moved to the methodology section. |
I have completed the introduction with references to sustainability of education. The line from 49 to 52 were moved in the methodology section. |
The conceptual model of Figure 1 should largely be described, and linked to hypotheses. |
I added a paragraph explaining how the set of hypotheses was constructed based on previous literature and research. I redrawed the diagram, adding H5 and another missing hypothesis. I have also labeled the hypothesis on the diagram
|
I am missing the indicators for each dimension in page 7. |
I added a table in the Appendix A to show the initial model, with all the indicators for each dimension. |
For numbers, use appropriate SI units, with points for decimals. |
I have made all replacements in the article according to International System of Units |
Rephrase ‘meaning that my correlation matrix of coefficients is not an identity matrix.’. The correlation matrix is not yours. Correct also ‘I can say that my data are sufficient and suitable for factor analysis.’ You cannot say: my data. Whenever possible, do not use the possessive ‘my’: my model, my data, etc… |
Changes have been made according to your suggestions throughout all article |
In the discussion, remove: As I pointed out in the theoretical part of this article. The discussion is poor and should be fully referenced with references from the educational world. In the whole manuscript I am missing references on sustainability. |
I have removed the indicated part of text. The discussion section was supplemented with an additional analysis of two of the open-ended questions in the questionnaire; references have been added to the sustainability of education with implications for sustainability of education |
This study should include a final paragraph on limitations. |
The final paragraph which refers to limitations of the study has been expanded |
Thank you for your time and for your help to improve this article!
Sincerely,
Ciprian Obrad, West University of Timișoara, Department of Sociology

Round 2
Reviewer 3 Report
Dear authors, thanks for accepting my comments. I have still some minor considerations:
I am suggesting a title change: "Constraints of online education, resilience and implications for sustainability." At least it is not necessary to introduce the factor 'Romanian'.
Line 50: Please, define the SD4
Line 387: Rephrase by not using 'my model'
Yours sincerely
Professor JC
Author Response
I made changes recommended by you as follows:
Comments and Suggestions |
Detalis of the revisions |
I am suggesting a title change: "Constraints of online education, resilience and implications for sustainability." At least it is not necessary to introduce the factor 'Romanian'. |
I have changed the title as you suggested |
Please, define the SD4 |
I have defined SDG4 at indicated lines |
Rephrase by not using 'my model' |
I rephrased without “my”
|
Thank you for your time and for your help to improve this article!
Sincerely,
Ciprian Obrad, West University of Timișoara, Department of Sociology

This manuscript is a resubmission of an earlier submission. The following is a list of the peer review reports and author responses from that submission.